# Why LLMs Hallucinate, And How To Get (Evidential) Closure: Perceptual, Intensional and Extensional Learning for Faithful Natural Language Generation

**Adam Bouyamourn**
UC Berkeley
adam.bouyamourn@berkeley.edu

## Abstract

We show that LLMs hallucinate because their output is not constrained to be synonymous with claims for which they have evidence: a condition that we call *evidential closure*. Information about the truth or falsity of sentences is not statistically identified in the standard neural language generation setup, and so cannot be conditioned on to generate new strings. We then show how to constrain LLMs to produce output that satisfies evidential closure. A multimodal LLM must learn about the external world (perceptual learning); it must learn a mapping from strings to states of the world (extensional learning); and, to achieve fluency when generalizing beyond a body of evidence, it must learn mappings from strings to their synonyms (intensional learning). The output of a unimodal LLM must be synonymous with strings in a validated evidence set. Finally, we present a heuristic procedure, *Learn-Babble-Prune*, that yields faithful output from an LLM by rejecting output that is not synonymous with claims for which the LLM has evidence.

## 1 Introduction

There is a growing body of evidence that LLMs systematically hallucinate (Ji et al., 2023; Maynez et al., 2020; Bang et al., 2023; Guerreiro et al., 2023; Dale et al., 2023). Hallucinations may limit the utility of LLMs, in addition to having significant implications for safety (Müller et al., 2020; Martindale and Carpuat, 2018; Martindale et al., 2019; Bender et al., 2021).

It has been suggested that hallucinations occur because language models do not interpret training inputs *semantically* (Bender and Koller, 2020; Xiao and Wang, 2021; McKenna et al., 2023). We offer a new formalization of this notion that allows us to explain why LLMs are inherently prone to hallucination, and what any faithful LLM must do: its output must be *closed under synonymy with its evidence about the world*, a condition we call

**evidential closure**. An LLM is *factual* if it is faithful, and, in addition, its evidence about the world is correct.[1]

Many of the conceptual issues now studied in natural language processing have received extensive treatment in the analytic philosophy of language (Quine, 1960; Davidson and Harman, 1972; Evans, 1982; McFetridge et al., 1992; McDowell, 1998). Conveniently, these treatments are often mathematically tractable.

One set of fundamental distinctions is between *intension* or *meaning*; *extension* or *reference*; and *facts* or *states of the world*, respectively.[2] Words and sentences have *meanings*, which are equivalence classes of other words or sentences with which they are synonymous. They also have *referents*: states of the world that they map onto. Finally, there is an *external reality* that the agent has access to, equipped with a valuation function that assigns states of the world to true or false. Sentences are true when they correctly refer to states of the world that are true.

A popular theory of meaning in the philosophy of language that links these three notions is the extensional semantics of Davidson (1967). This theory holds that *the meaning of a sentence is just the set of states of the world in which that sentence is true*.

Using this account, we can characterize a *faithful* speaker of a language as one who 1) uses their knowledge about the world 2) to describe states of the world 3) using a variety of equivalent sentences. This entails that a faithful speaker must perform three tasks: they must learn about the world

---

[1]We study *intrinsic* (Huang et al., 2023), or *input-conflicting* hallucinations (Zhang et al., 2023); and *extrinsic*, or *fact-conflicting* hallucinations. This conception of hallucination does not include all conceptions of hallucination in the literature: LLMs may produce output that is ill-formed or contextually irrelevant, for instance (Guerreiro et al., 2023).

[2]For accessible overviews, see Fitting (2022), Michaelson and Reimer (2022), David (2022).

(**perceptual learning**); they must learn which sentences map onto which states of the world (**extensional learning**); and they must learn which sentences have the same meaning (**intensional learning**). A *factual* speaker performs the same tasks, with the difference that their evidence about the world is correct. Here, faithfulness to model inputs is conceptually prior to factuality, since, definitionally, the information the model has about the world is contained in its inputs.

We use this setup to state an impossibility result: neural probabilistic language models in the vein of Bengio et al. (2003) are not factual (Theorem 4.5). LLMs maximize the conditional probability of the generated strings given the corpus and the prompt. They do not explicitly learn states of the world, do not explicitly learn the meanings of words, and do not explicitly learn the referents of sentences. Each of these types of information is unobserved. As a result, the conditional distributions learned by LLMs can be statistically independent of, or invariant to, their semantic content: that is, of the referents and the truth-conditions of the sentences in the corpus and prompts. So we may have variation in the truth or falsity of a state of the world without variation in the solution to a predictive model that generates the next sentence given a prompt.

Because this semantic information is not contained in the conditional distribution from which an output string is generated, simulating from the learned distribution does not preserve this information, *even when it is contained in the corpus* (Theorem 4.6). Hence there is no guarantee that LLMs are faithful to the semantic content of their inputs, either. We can think of this as the *cause* of hallucination in LLMs.

Second, we show conceptually how to build a faithful or factual LLM. The output of such an LLM must satisfy *evidential closure*: that is, its output must be synonymous with claims for which the LLM has evidence. This ensures that every claim made by the model is either directly corroborated by evidence, or is a paraphrase of a directly corroborated claim. We first define the objective functions for faithful and factual LLMs using theory from the philosophy of language. We then decompose those objective functions into learnable distributions (Model 5.9). We show that the output of these models is faithful or factual, because it solves the constrained learning task that incorporates semantic information about the truth or fal-

sity of sentences for which the model has evidence (Theorem 5.10 and Theorem 5.11).

Third, we provide heuristic framework for building factual or faithful LLMs, which we call **Learn-Babble-Prune**. In this setup, the output of an LLM is cross-checked against its evidence, and discarded if it is not a paraphrase of a claim that it has evidence for. This ensures that the output of an LLM is consistent with its evidence. This ensures that the LLM is faithful *to its evidence*, and hence, does not hallucinate.

We consider two applications: when the evidence is a corpus of strings, and when the evidence is in the form of sensor information.

If the evidence is in the form of text, as in some retrieval-augmented language models (Guu et al., 2020; Chen et al., 2017), then an LLM's output is consistent with its retrieved body of evidence *if its output is a paraphrase of a string in its evidence base*. In this application, the model must learn paraphrases of sentences in order to cross-check the output of an LLM with its evidence base.[3]

If the evidence is in the form of information about the world gathered by sensors, as in some multimodal LLMs (Lian et al., 2023; Zhao et al., 2023b; Yuan et al., 2022), then, in addition to intensional learning, the LLM must learn a mapping from perceptual information to strings. Thus a multimodal LLM is faithful if its output paraphrases information acquired by its perceptual learner, and factual if that perceptual learner is unbiased and consistent.

A final point is that, like any language speaker, an LLM can only make as many true claims as are semantically entailed by the evidence it possesses (Williamson, 2000). Collecting and interpreting large bodies of evidence is a vitally important task for the next generation of LLMs.

## 2 Related work

### 2.1 Retrieval-Augmented Generation

Several approaches ground LLMs in external textual knowledge bases (Chen et al., 2017; Lee et al., 2019; Guu et al., 2020), in which a retriever model

---

[3]Intensional learning allows the speaker to say true things that they have never heard before: given a candidate string $\ell$, the speaker can verify it by first verifying the state of the world underpinning a different string $\ell^+$, and then verifying that $\ell$ is intensionally equivalent to $\ell^+$. This is an example of *semantic entailment* (Beth, 1955). A fluent speaker of a language must be able to generalize beyond a set of observed sentence-use instances, and intensional learning allows speakers to do this.

is trained in order to learn subsets of the training data that are more relevant for conditional text generation. Asai et al. (2022) and Lee et al. (2021) highlight the role of evidence in controlling incidence of hallucination: models are less likely to hallucinate if they focus attention on passages with a high evidential value. Our approach provides theory to formalize the notion of evidential relevance, and proposes ex post consistency with the evidence to enforce the factuality or faithfulness of the output of an LLM.

## 2.2 Generalizable RL

Methods for generalizing reinforcement learning models in complex environments are also closely related (Agarwal et al., 2021; Zhao et al., 2023b; Belyaeva et al., 2023; Lian et al., 2023). We can understand differences in semantic content as a form of distribution shift, in which the desired output distribution and the observed input distribution differ. Hence, generalizing across contexts in an RL setting is analogous to the problem of factual language generation: in each case, the output must be conditioned on the correct semantic information. This is not possible when that information is not observed.[4]

## 2.3 Causal Inference

The ability to generalize across semantic contexts is also closely connected to the identifiability of semantic information. Identifiability is a well-studied problem in causal inference in both computer science (Pearl, 1995; Shpitser and Pearl, 2012; Bareinboim and Pearl, 2012; Lee and Bareinboim, 2020; Li et al., 2023) and in statistics, epidemiology, and the social sciences (Holland, 1986; Petersen and van der Laan, 2014; Lewbel, 2019). One moral from this paper is that a grounded LLM is one that is causally connected to its environment. This also has a philosophical foundation in causal theories of perception (Grice and White, 1961; Hyman, 1992).

## 3 Setup and theory

### 3.1 A very brief philosophical overview

Aristotle (350 BCE [1989]), in *Metaphysics*, gave the following definition of truth: "To say of what

is that it is, or of what is not that it is not." This is a correspondence theory of truth, in which truth is a property of sentences that correspond to states of the world (Schmitt, 2003).

Davidson (1967), building on work by Tarski (1936), equated the meaning of a sentence with the circumstances in which that sentence was true: that the *meaning* of the sentence "$p$" is just that the state of the world $p$ obtains. This is an *extensional semantics*: that the meaning of a sentence *is* the circumstances in which it is true.

Frege (1892) noticed that distinct terms in a language with the same referent could have different meanings associated with them. He described the *meaning* (intension) and *reference* (extension) of a term, noticing that it could count as nontrivial knowledge for a person to note that "the Morning Star", and "the Evening Star", two terms with seemingly different senses in fact referred to the same object, Venus. It is one type of knowledge to learn about objects; it is a different type of knowledge to learn about the *names* of objects. To a language user, *learning about intensions* is different from *learning about extensions*. As Quine (2008) put it: "truth depends in general on language and fact."

Learning about the world, learning which words refer to which states of the world, and learning which words are synonymous together allow a speaker to attain fluency, by making true statements beyond a finite phrasebook of verified claims. A speaker whose utterances are constrained to be consistent with their evidence remains faithful to their knowledge about the world.

We now formalize these ideas.

### 3.2 Formalization

Consider a stock of syntactically well-formed strings of a natural language $\ell \in L$; a set of possible states of the world $\omega \in \Omega$; and a set of extensional or reference maps $R : L \to \Omega$, which map strings onto states of the world. We write $R(\ell) = \omega$, to show that $\ell$ is the string of natural language that has as its referent the state $\omega$. We say that the *pair* $\langle \ell, R(\ell) \rangle$ is a *sentence*, or *interpreted string*.

We require that the language $L$ is *fully-interpreted* and *unambiguous*: for each sentence $\ell \in L$, there exactly one element of $\omega \in \Omega$ such that $R(\ell) = \omega$. We also require that the domain $\Omega$ is non-empty.

A *semantic structure* $s \in S$ assigns to each state of the world $\Omega$ a binary value. We associate

---

[4]Agarwal et al. (2021) point out that generalization failures can happen even when the input and the output contexts are the same context. This is because, even though the contexts are identical, the model did not learn the relevant semantic information in each context, so its output is not conditioned on the relevant information. See Section 4.4.

each distinct structure $s$ with a unique mapping $V_s : \Omega \to \{0, 1\}$, which we call a *valuation function*. Each structure $s$ represents a possible assignment of values to states of the world.

We are interested in pairs $\langle \ell, R(\ell) \rangle$ evaluated under structures $s \in S$. That is, we are interested in strings and their referents – sentences – and the assignment of truth-values to those sentences.

We start with states of the world.

**Definition 3.1.** *A state of the world $\omega \in \Omega$ obtains under a structure $s$ if $V_s(\omega) = 1$.*

We take a *particular* structure $s_0$ as the one that describes the actual world, and denote its valuation function $V_0$. We say that a state of the world $\omega$ *actually obtains* if $V_0(\omega) = 1$. Learning about the real world therefore involves learning which facts *actually obtain*, or $\forall \omega : V_0(\omega)$.[5]

We next describe reference. The extensional map $R : L \to \Omega$ maps strings of a natural language onto states of the world. For example, the sentence "Joe Biden won the 2020 President election" is a string that refers to the state of the world in which Joe Biden won the 2020 Presidential election.

**Definition 3.2** (Reference/Extension)**.** *A string $\ell$ refers to, has as its extension, or describes $\omega \in \Omega$, if $R(\ell) = \omega$.*

Truth is a property of strings and the states of the world that they describe. If the sentence successfully describes a state of the world that in fact obtains, we say that the sentence is true.

**Definition 3.3** (Truth)**.** *A sentence $\langle \ell, R(\ell) \rangle$ is true under a structure $s$ if and only if the state of the world it describes obtains under $s$, that is, $V_s[R(\ell)] = 1$.*

We are specifically interested in the privileged structure that describes reality, $s_0$. That is, we are interested in sentences that are *actually true*, because they refer to states of the world that actually obtain.

The meaning of a sentence, in Davidson's account, is just the set of circumstances in which a sentence is true. We say that two sentences are *synonymous*, or *intensionally equivalent*, if they are true on all and only the same structures.

**Definition 3.4** (Synonymy/Intensional Equivalence (Davidson))**.** *Sentences $\langle \ell_i, R(\ell_i) \rangle$, $\langle \ell_j, R(\ell_j) \rangle$ are synonymous, or intensionally equivalent, if $\forall s : V_s[R(\ell_i)] = V_s[R(\ell_j)]$.*

---

[5] We use the word 'obtains' to distinguish valuations on states of the world from valuations on sentences.

Equivalently, two sentences are synonymous if there exists no assignment of values to states of the world that would make one sentence true and the other false. For example, "The Eiffel Tower is the tallest building in France", and "It is not the case that the Eiffel Tower is not the tallest building in France" are true in all and only the same circumstances – they are synonymous. There is no set of possible assignments of values to states of the world that would make one of these claims true and the other false.

Because meaning, on this account, is defined in terms of truth, synonymy is *truth-preserving*. That is, if we know that a particular sentence is true, then we know that any sentence synonymous with that sentence is also true.

**Proposition 3.5** (Closure under synonymy)**.** $\forall \ell, \ell' \in L : V_0[R(\ell)] = 1$ *and* $R(\ell) = R(\ell') \implies V_0[R(\ell')] = 1$

*Proof.* Apply Definition 3.4. $\square$

Proposition 3.5 says that if we start with a sentence that is true, then any sentence that has the same meaning as that sentence is also true. This is important for allowing language models to make claims beyond an existing knowledge base of verified claims: to ensure than an LLM is faithful, its output must *be closed under synonymy with its evidence*.

## 4 Why LLMs Hallucinate

### 4.1 Setup

Consider a neural language generation task, as motivated by Bengio et al. (2003) and others. The analyst observes a training corpus of strings $C = \{x_i\}_{i=1}^n$, and learns a (possibly masked) model that maximizes:

$$\hat{f}(C) = \prod_i \arg\max_{x_i} Pr(x_i | \boldsymbol{x}_{i' \neq i}) \quad (1)$$

Then, given a prompt $P$, the model generates an output string $\hat{y}$:

$$\hat{y} = \arg\max_x \hat{f}(x | C, P) \quad (2)$$

Good out-of-sample performance on conventional benchmarks is taken as evidence that $\hat{f}(x | C, P) \approx f(x | P)$ (Zhao et al., 2023a).

## 4.2 Characterizing Factual and Faithful LLMs

A factual LLM is one that produces *only true sentences*.

**Definition 4.1** (Factual LLMs). *An LLM is factual if* $\forall \hat{y} : V_0[R(\hat{y})] = 1$.

That is, an LLM is factual if every string output of the LLM refers to a state of the world that actually obtains.

A factual LLM therefore solves (using Definition 3.3):

**Problem 4.2** (Factual LLMs).

$$\max_x f(x|C, P) \ s.t. \ V_0[R(x)] = 1$$

This constraint encodes two additional pieces of information: what state of the world the output sentence $\hat{y}$ refers to, and whether or not it obtains.

A *faithful* LLM is one that produces sentences that are semantically entailed by the agent's evidence. Suppose that we have an estimator of $\hat{V}_0$. Then,

**Definition 4.3** (Faithful LLMs). *An LLM is faithful if* $\forall \hat{y} : \hat{V}_0[R(\hat{y})] = 1$.

A faithful LLM solves:

**Problem 4.4** (Faithful LLMs).

$$\max_x f(x|C, P) \ s.t. \ \hat{V}_0[R(x)] = 1$$

Here the constraint is that the output is consistent with an estimated truth-value. If $\hat{V}_0$ is a biased estimator of $V_0$, consistency with the model's evidence does not guarantee that its output is true.

A natural way to state this is that an LLM is *faithful* if its output is *consistent with its information about the world*. An LLM is *factual*, if, in addition, that information is accurate. In this formalization, we can say that a faithful LLM is factual if $\hat{V}_0 = V_0$.[6]

## 4.3 Truth as an unidentified nuisance parameter

The solution to Problem 4.2 depends on information that is not learned in the setup of Equation (2). This leads to an *identification problem* (van der

---

[6]In practice, we might be interested in different asymptotic conceptions of factuality: for instance, an LLM could be almost surely factual if $\hat{V}_0 \overset{a.s.}{\to} V_0$.

Vaart, 1998, 62), in the sense that, for two possible structures $s, s'$, and for any given output string $\hat{y}$:

$$
\begin{aligned}
&V_s[R(\hat{y})] \neq V_{s'}[R(\hat{y})] \implies \\
&\arg\max_x \hat{f}(x|C, P, V_s[R(x)]) \neq \\
&\arg\max_x \hat{f}(x|C, P, V_{s'}[R(x)])
\end{aligned}
\tag{3}
$$

That is, we may have different assignments of truth-values to states of the world, without any difference in which sentence is generated by the LLM. Joe Biden in fact won the 2020 Presidential election, but given a particular prompt and corpus, "Donald Trump won the 2020 Presidential election" may be the sentence that has the highest conditional probability of being observed. This is because the language model does not observe the state of the world referred to by either string, and does not output a sentence conditional on this information.

Truth-values of states of the world are not observed or identified in the model. Hence we have $\forall s : \hat{f}(x|C, P, V_s) = \hat{f}(x|C, P)$. And in particular, it follows that, for every $s \neq s_0 : \hat{f}(x|C, P, V_s) = \hat{f}(x|C, P)$. The model solution is invariant to assignments of truth-values to states of the world. To put it another way, $V_0$ is an *unidentified nuisance parameter* (Basu, 1977). This entails that:

$$\arg\max_x \hat{f}(x|C, P) = \hat{y} \implies V_0[R(\hat{y})] = 1 \tag{4}$$

Or, in other words, a sentence may be the solution to the maximization problem even if it is false. And hence, there is no guarantee than an LLM will be strictly truthful. In general, for any $\epsilon > 0$:

$$
\begin{aligned}
&D_{KL}\{\hat{f}(x|C, P)||f(x|C, P)\} < \epsilon \implies \\
&D_{KL}\{\hat{f}(x|C, P, V_0)||f(x|C, P, V_0)\} < \epsilon
\end{aligned}
\tag{5}
$$

Statistical similarity of any two distributions does not imply statistical similarity of two distributions that depend on additional semantic information about the world.

## 4.4 A verified training corpus is not enough: Similarity does not entail synonymy

It might be hoped that a model of the type of Equation (2) solves Problem 4.2 *indirectly*. After all, doesn't the training corpus contain information about states of the world (Akyürek et al., 2022)? And don't LLMs seem to generate useful information, or correctly summarize existing information, *some* of the time (Petroni et al., 2019)?

Firstly, if the training corpus contains statements that are false, ambiguous, or fail to refer to any entity, it is straightforward to see that any set of generated sentences can also contain false, ambiguous statements, or exhibit referential failure.

We say that a training corpus is *verified* if it contains only true, unambiguous statements with no referential failure. But it is still possible for a model that solves Equation (2) to fail to solve Problem 4.2.

A major innovation in the LLM literature has been to create models that successfully generate sentences that have not been previously observed, via encoder-decoder representations of the space of sentences (Wu et al., 2016; Vaswani et al., 2017). Interestingly, however, this actually makes it *more* likely that LLMs will produce false sentences, since it makes it possible for the model to generate sentences that go beyond the information contained within the verified training corpus. If these previously-unseen sentences are not also constrained to refer to facts about the world, there is no guarantee that these will be true, even when the training corpus is verified.

The problem is that *similarity does not entail synonymy*: we have no guarantee that a generated sentence is synonymous with any sentence in the training corpus (see Proposition 3.5). Distribution shift from the training context to the desired output context is *always* possible when the LLM does not learn the correct distribution.

### 4.5 Formal results

**Theorem 4.5** (LLMs are not factual)**.**

$$\arg\max_x \hat{f}(x|C,P) = \hat{y} \;\not\Rightarrow\; V_0[R(\hat{y})] = 1$$

*Proof.* Consider structures $s, s'$ such that $V_s[R(\hat{y})] = 1$ and $V_{s'}[R(\hat{y})] = 0$. Since $V_s, V_{s'}$ are unobserved, we have that $\hat{f}(x|C,P,V_s) = \hat{f}(x|C,P,V_{s'}) = \hat{f}(x|C,P)$. Set $V_0 = V_{s'}$. Then $\arg\max_x \hat{f}(x|C,P) = \hat{y}$ but $V_0[R(\hat{y})] = 0$, and the claim follows. $\square$

**Theorem 4.6** (Training on verified information does not induce factuality)**.**

$$\arg\max_x \hat{f}(x|C,P) = \hat{y} \;\wedge$$
$$\forall x_i \in C : V_0[R(x_i)] = 1 \;\not\Rightarrow$$
$$V_0[R(\hat{y})] = 1$$

*Proof.* Suppose $\forall x_i \in C, R(\hat{y}) \neq R(x_i)$, and consider the structures $s, s'$ such that $\forall x_i \in C :$ $V_s[R(x_i)] = V_{s'}[R(x_i)] = V_0[R(x_i)] = 1$. Suppose $\exists x' \notin C, V_s[R(x')] = 1$ and $V_{s'}[R(x')] = 0$. Then take $\hat{y} = x'$ and $V_0 = V_{s'}$, and the claim follows. $\square$

The key step in each argument is that no information about $V_0$ is learned by the model. So a structure can always exist on which the output sentence is false. And since we do not observe the truth-values of states of the world, we have no way of ruling out the possibility that the structure on which the sentence is false is in fact $s_0$.

This result does not say that LLMs *always* hallucinate. But it does say that, when an LLM learns a distribution that does not incorporate explicit factual or extensional information, it is always *possible* for an LLM to hallucinate.

## 5 Building Factual/Faithful LLMs: How To Get (Evidential) Closure

How do we go beyond the negative result above? We require that the output of an LLM is equal to *the closure under synonymy of strings that refer to verified information*. That is, every string output by the LLM either refers to a claim that is verified, or is a synonym of a claim that is verified. By Proposition 3.5, all such sentences will be true. Any LLM that satisfies this property is then factual. If we require only that the output is closed under synonymy with strings that are synonymous with the agent's evidence, irrespective of its credibility, the LLM is faithful.

### 5.1 A Symmetry Group Semantics

It is helpful to use the technology of symmetry groups to formally motivate intensional and extensional learning. This section restates and develops results in Kiddon and Domingos (2005).

**Definition 5.1** (Symmetry)**.** *A function $g : X \to X$ is a symmetry of a set $X$ if $\{g(x) \mid x \in X\} = X$.*

**Definition 5.2** (Symmetry Group)**.** *A symmetry group of a set $X$ is an ordered pair $(G, \circ)$ such that if $g$ is a symmetry of $X$ then $g \in G$, and $\circ$ is function composition.*

**Definition 5.3** (Orbit)**.** *The orbit of $x \in X$ under a symmetry group $G$ is the set $\{g(x) \mid g \in G\}$*

We motivate paraphrases as functions that, given a list of strings, permute pairs of strings that have the same referent.

**Definition 5.4** (Paraphrase Map). *A bijective function* $\pi : L \to L$ *is a paraphrase map if* $\forall \ell : R(\ell) = R(\pi(\ell))$

Essentially, $\pi$ is a permutation, with the added constraint that it can permute only strings in a list that have the same referent. We collect the set of paraphrases in the collection $\Pi$. This is a symmetry group of $L$, since, every $\pi$ in $\Pi$ is a permutation of the elements of $L$, and hence applying $\pi$ to $L$ returns the same list of strings, that is, $L$.

**Proposition 5.5.** *The set of paraphrase maps* $(\Pi, \circ)$ *is a symmetry group of the set $L$.*

*Proof.* We suppose that that $\Omega$ is non-empty, and that $L$ is fully-interpreted and unambiguous, so that, for each $\ell \in L, \exists! \, \omega \in \Omega : R(\ell) = \omega$. Then, since each $\pi \in \Pi$ is a bijection, and hence is a permutation, we have that, $\forall \pi \in \Pi, \{\pi(\ell) : \ell \in L\} = L$, so that each $\pi$ is a symmetry of $L$. It is straightforward to show that $(\Pi, \circ)$ satisfies the group axioms: any composition of permutations $\pi, \pi' \in \Pi$ defines a permutation $\pi'' \in \Pi$; composition is associative; there is a trivial permutation; and each permutation has an inverse permutation. Hence $(\Pi, \circ)$ is a symmetry group of the set $L$. $\qquad\square$

**Definition 5.6** (Semantic Orbit). *The orbit of* $\ell \in L$ *under* $\Pi$ *is the set* $I(\ell) = \{\pi(\ell) | \pi \in \Pi\}$.

That is, the semantic orbit of a sentence is the set of sentences that refer to the same state of the world as that sentence.[7] We collect the set of unique orbits of a language $L$ in the set $\mathscr{I}$.

### 5.2 Factual LLMs

With this setup in place, we are now in a position to decompose the constrained learning task described in Problem 4.2. We introduce a *source* language $L^+$, which contains the strings in some source set of sentences, and $\mathscr{I}^+$ the set of unique orbits of $L^+$. We can then rewrite the objective function as follows:

---

[7]Note that the set of unique orbits of $L$ partitions $L$ into sets of strings that have the same referent. Further, for each unique orbit, there is a unique state of the world to which every sentence in that orbit refers. We can extend the notation of a valuation function so that it takes the set of unique orbits as its pre-image. We write: $V_0[I(\ell)] = 1 \iff \forall \ell \in I(\ell) : V_0[R(\ell)] = 1$.

**Proposition 5.7.**

$$f(\ell | C, P, V_0[R(\ell)] = 1)$$
$$= \sum_{I \in \mathscr{I}^+} f(\ell | C, P, \ell \in I(\ell^+) \cap V_0[I(\ell^+)] = 1)$$
$$= \sum_{I \in \mathscr{I}^+} \underbrace{f(\ell | C, P)}_{\text{LLM}} \underbrace{f(\ell | I(\ell^+))}_{\substack{\text{Intensional} \\ \text{Learner}}} \underbrace{f(V_0[I(\ell^+)] = 1)}_{\substack{\text{Ground truth} \\ \text{or Evidence}}}$$

Here, $f(V_0[I(\ell^+)] = 1)$ represents the information about the world in each string $\ell^+$ of the source language $L^+$.[8]

If we do not have a ground truth set of strings, but instead learn about the world via sensors (a vision model, for example), we can further decompose $f(V_0[I(\ell^+)] = 1)$ as follows:

**Proposition 5.8.**

$$f(V_0[I(\ell^+)] = 1)$$
$$= f(R[I(\ell^+)] = \omega \cap V_0(\omega) = 1)$$
$$= \sum_{\omega \in \Omega} f(I(\ell^+) | V_0(\omega) = 1) Pr(V_0(\omega) = 1)$$
$$= \sum_{\omega \in \Omega} \underbrace{f(I(\ell^+) | \omega)}_{\substack{\text{Extensional} \\ \text{Learner}}} \underbrace{f(\omega)}_{\substack{\text{Perceptual} \\ \text{Learner}}}$$

**Model 5.9** (A Factual/Faithful Multimodal LLM).

$$f(\ell | C, P, V_0[R(\ell)] = 1) =$$
$$\sum_{I \in \mathscr{I}^+} \sum_{\omega \in \Omega} \underbrace{f(\ell | C, P)}_{\text{LLM}} \underbrace{f(\ell | I(\ell^+))}_{\substack{\text{Intensional} \\ \text{Learner}}} \underbrace{f(I(\ell^+) | \omega)}_{\substack{\text{Extensional} \\ \text{Learner}}} \underbrace{f(\omega)}_{\substack{\text{Perceptual} \\ \text{Learner}}}$$

In words, this models the constraint that $\omega$ obtains, there is a sentence in the source language that refers to $\omega$, and that the output sentence $\ell$ is synonymous with a sentence in the source language. Hence Model 5.9 satisfies evidential closure. The representation in Model 5.9 above covers both factual and faithful LLMs.

**Theorem 5.10.** *Suppose* $\hat{f}(\omega)$ *is an oracle perceptual learner, we have consistent intensional and extensional learners, and an LLM that solves Equation* (2). *Then Model 5.9 is factual.*

*Proof.* We have shown in Proposition 5.7 and Proposition 5.8 that $f(\ell | C, P, V_0[R(\ell)] = 1) = f(\ell | C, P) f(\ell | I(\ell^+)) f(I(\ell^+) | \omega) f(\omega)$. Hence

---

[8]Encyclopedia authors have essentially already done the extensional and perceptual learning for us: they have encoded information about the environment as text. It is in this sense that perceptual and extensional learning are necessary for truthful language generation even in the textual case.

Problem 4.2 is solved if and only if this model can be consistently estimated. Since $\hat{f}(\omega)$ is an oracle, $V_0(\omega) = 1 \iff \hat{f}(\omega) = 1$, and since $\hat{f}(\ell|I(\ell^+))\hat{f}(I(\ell^+)|\omega) \to f(\ell|I(\ell^+))f(I(\ell^+)|\omega)$ by assumption, we have that $\hat{f}(\ell|C,P)\hat{f}(\ell|I(\ell^+))\hat{f}(I(\ell^+)|\omega)\hat{f}(\omega) \to f(\ell|C,P,V_0[R(\ell)] = 1)$, solving Problem 4.2. $\square$

**Theorem 5.11.** *If we have a perceptual learner that is not an oracle, consistent intensional and extensional learners, and an LLM that solves Equation (2), then Model 5.9 is faithful to its perceptual learner.*

*Proof.* Define $\hat{V}_0(\omega) \equiv \hat{f}(\omega)$. Then, $\hat{V}_0(\omega) = 1 \iff \hat{f}(\omega) = 1$, and $\hat{f}(\ell|C,P)\hat{f}(\ell|I(\ell^+))\hat{f}(I(\ell^+)|\omega)\hat{f}(\omega) \to f(\ell|C,P,\hat{V}_0[R(\ell)] = 1)$, solving Problem 4.4. $\square$

### 5.3 Evidence Set Representation

We can state these results even more simply, however. Consider the following set:

**Definition 5.12** (Evidence Set)**.**

$$\hat{E} \equiv \{\ell|\hat{V}_0[\hat{I}(\ell)] = 1\}$$

We say that $\hat{I}$ is the output of an intensional learner: the set of learned paraphrases of strings. $\hat{V}_0$ is the set of stipulated or learned labels attached to strings and their paraphrases. In either case, this is the set of strings that comprise, or are synonymous with, a body of verified information about the world.

This definition is helpful, because it allows us to re-express Model 5.9 as:

**Model 5.13** (An Evidence-Grounded LLM)**.**

$$f(\ell|C,P)\mathbb{I}\{\ell \in \hat{E}\}$$

**Theorem 5.14.** *Suppose $\hat{I} \to I$. Then Model 5.13 is faithful.*

*Proof.* $\hat{I} \to I \implies \mathbb{I}\{\ell \in \hat{E}\} \to 1 \iff \hat{V}_0[R(\ell)] = 1$, so Model 5.13 solves Problem 4.4. $\square$

**Theorem 5.15.** *Suppose $\hat{I} \to I$ and $\hat{V}_0 \to V_0$. Then Model 5.13 is factual.*

*Proof.* $\hat{V}_0[\hat{I}(\ell)] \to 1 \implies V_0[\hat{I}(\ell)] = 1$, and $\hat{I} \to I \implies \mathbb{I}\{\ell \in \hat{E}\} \to 1 \iff V_0[I(\ell)] = 1$, so Model 5.13 solves Problem 4.2. $\square$

$\hat{E}$ denotes the set of strings that consist of both model's explicit evidence about the world, and their paraphrases. This set is evidentially closed, by Proposition 3.5. So the output of Model 5.13 is evidentially closed, and the model is faithful to its evidence.

The moral is that a faithful or factual LLM must learn about the world, directly (through sensor perception) or indirectly (through text corpora); and that its output must be constrained to be synonymous with claims that are contained within its evidence set. This provides theoretical motivation for grounding, and clarifies specifically what grounding is intended to accomplish.

### 5.4 Learn-Babble-Prune: A framework for factual/faithful LLMs via rejection sampling

We propose a procedure we call *Learn-Babble-Prune*[9] to implement this.

In the *Learn* phase, an agent learns about the world, either directly through perceptual learning, or indirectly by observing some stock of verified information. If this information is acquired through perception, these sentences are translated into natural language, via multimodal encoder-decoder system. The agent additionally learns a stock of paraphrases of sentences, via paraphrase corpora methods (Ormazabal et al., 2022), or via contrastive learning methods (Yang et al., 2021).

In the *Babble* phase, an LLM generates a stock of candidate sentences.

In the *Prune* phase, a generated sentence is cross-checked against the its Evidence Set. If the sentence is verified, or if it is a paraphrase of a verified sentence, it is printed. Otherwise, it is rejected.

### 5.5 Applications of Learn-Babble-Prune

#### 5.5.1 Example 1: Text-To-Text

**Learn** *Ground truth.* Scrape an online encyclopedia, and designate this as the Evidence Set $\hat{E}$.
*Intensional learning.* Learn a set of paraphrases of strings in the Evidence Set, and add them to the Evidence Set.
**Babble** An LLM generates a response to a query.
**Prune** The response is rejected if it is not a paraphrase of a sentence in the Evidence Set.

---

[9]This was inspired by He (2018).

### 5.5.2 Example 2: Image-To-Text

**Learn** *Extensional learning.* Pre-train a visual encoder, which learns a mapping from images (states of the world) to strings.
*Perceptual learning.* Designate a test set of images as ground truth about the environment. Apply the visual encoder to this test set of images. Designate its output as our Evidence Set.
*Intensional learning.* Learn a set of paraphrases of strings in the Evidence Set, and add them to the Evidence Set.
**Babble** An LLM generates a response to a query.
**Prune** The response is rejected if it is not a paraphrase of a sentence in the Evidence Set.

---

**Algorithm 1** LBP: Text-to-Text

---

1: Input: $(L, \hat{E}, C, P)$
2: Learn: $\hat{f}(x|C, P)$                    ▷ Learn
3: Learn: $\forall \ell^+ \in \hat{E}, \forall \ell \in L: \hat{f}(\ell \in I(\ell^+))$
4: **for** $\ell \in L$ **do**
5:     **if** $\ell \in I(\ell^+) \wedge \ell^+ \in \hat{E}$ **then**
6:         $\hat{E} \leftarrow \hat{E} \cup \ell$
7:     **end if**
8: **end for**
9: Generate: $\hat{y} \sim \hat{f}(x|C, P)$              ▷ Babble
10: **if** $\hat{y} \in \hat{E}$ **then**                    ▷ Prune
11:     Print: $\hat{y}$
12: **else**
13:     Print: "I don't know."
14: **end if**

---

Since the source domain is arbitrary, Example 2 covers a wide variety of use cases, which can of course be combined. The output of these procedures is faithful, because if a given candidate output is not synonymous with a claim for which the model has have explicit evidence, it is not printed.[10]

### 5.6 The Limits of Factual or Faithful LLMs

Any model of the type of Model 5.13 is limited in what it can say by the size of its evidence base. In practice, the dimension of $\hat{E}$ may be considerably smaller than the parametric knowledge of the language stored in many LLMs. Any use-case for factual LLMs requires the collection and verification of a large amount of factual information. Any factual or faithful LLM can only generate as much output as it can verify.

---

[10] Wittgenstein (1922, 189): "Whereof one cannot speak, thereof one must remain silent." This also applies to LLMs.

## 6 Conclusions

LLMs hallucinate because their output is not constrained to be semantically consistent with their inputs, so there is no guarantee that any evidence about the world contained in their inputs is preserved in their outputs.

To build a faithful or factual LLM it is necessary to constrain the output of a model to be consistent with claims for which the model has explicit evidence.

In practice, this means acquiring large bodies of machine-readable string evidence or using sensors (perceptual learning) combined with sensor-to-text encoders (extensional learning) to validate the output of an LLM with evidence. Paraphrase learning methods can be used to expand the LLM's vocabulary (intensional learning). We propose a simple method to implement this in practice via rejection sampling.

Any input-faithful LLM is limited in what it can say by what it has evidence for. Generating large-scale evidence bases is likely to be a much bigger binding constraint that the parameter size of the model, for instance, and may require a rethink of how computational and financial resources are allocated to the design of LLMs. This is a challenge for the next generation of LLMs.

---

**Algorithm 2** LBP: Multimodal-to-Text

---

1: Input: $(\Omega^{obs}, L, L^+, C, P)$
2: Learn: $\hat{f}(x|C, P)$                        ▷ LLM
3: Learn: $\hat{f}(\omega^{obs})$                   ▷ Perceptual
4: Learn: $\forall \ell^+ \in L^+: \hat{f}(\ell^+|\omega^{obs})$   ▷ Extensional
5: Learn: $\forall \ell \in L: \hat{f}(\ell \in I(\ell^+))$    ▷ Intensional
6: **for** $\ell^+ \in L^+$ **do**
7:     **if** $f(\ell^+|\omega^{obs})f(\omega^{obs}) = 1$ **then**
8:         $\hat{E} \leftarrow \hat{E} \cup \ell^+$
9:     **end if**
10: **end for**
11: **for** $\ell \in L$ **do**
12:     **if** $\ell \in I(\ell^+) \wedge \ell^+ \in \hat{E}$ **then**
13:         $\hat{E} \leftarrow \hat{E} \cup \ell$
14:     **end if**
15: **end for**
16: Generate: $\hat{y} \sim \hat{f}(x|C, P)$              ▷ Babble
17: **if** $\hat{y} \in \hat{E}$ **then**                    ▷ Prune
18:     Print: $\hat{y}$
19: **else**
20:     Print: "I don't know."
21: **end if**

---

## Limitations

### Empirical performance and truthfulness

Enforcing consistency with evidence via rejection sampling is a relatively inefficient way to constrain output to be factual or faithful. We expect that RL approaches could implement a procedure analogous to Learn-Babble-Prune more efficiently, with tolerable loss of accuracy. The purpose of this framework is to highlight, at a high-level, the kind of tasks that any such RL approach would have to do. As such, it is primarily intended as a conceptual contribution. Further, strict factuality may be an undesirably high bar in some applications: clearly it depends on the use case.

### Safety and ground truth

A factual LLM constructed as above would require either a gold-standard data set of labelled data. However, there are clear safety concerns raised by treating some data sets as ground truth compared to others. Further, what counts as evidence in some domains is contestable: reasonable people disagree. Widely available benchmark data sets have well-studied biases (Paullada et al., 2021).

### Compositionality

Our framework does not consider the semantic content of constituents of sentences, instead considering strings as primitive, and assuming that they each refer to one fact. It is straightforward to extend our account to sentences that refer to the composition of multiple states of the world and logical operators, which we leave to future work.

### Language and Vagueness

We assume that a stock of strings is interpretable and unambiguous. Many sentences in natural language cannot be considered to have a truth-value, or may be ambiguous.

### Philosophy of language

The relevant philosophy of language literature is vast and we cannot hope to do it justice in this paper. Further, some conceptual distinctions that are important to understanding the papers cited are not made in this paper. The hope is that the setting provides offers a statistically tractable implementation of conceptual material that is covered in the papers cited. Subtleties, and perhaps even major distinctions, are likely to be lost in translation.

## Acknowledgements

This paper is dedicated to Stephen Williams. The author would like to thank Micah Carroll, Kirk Bansak, Orr Paradise, and three anonymous referees for helpful comments.

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
