# OpenReview forum: "Why LLMs Hallucinate, and How to Get (Evidential) Closure: Perceptual, Intensional, and Extensional Learning for Faithful Natural Language Generation"
_EMNLP/2023/Conference — EMNLP 2023 Main_

### Official Review · Reviewer_gLhZ · 2023-07-27

**Soundness:** 4

**Excitement:**

4: Strong: This paper deepens the understanding of some phenomenon or lowers the barriers to an existing research direction.

**Paper Topic And Main Contributions:**

The paper provides a truth-teoretic account on why LLMs allucinate. It further establishes the logical preconditions that must be met for a language model not to allucinate. These pre-conditions are that a truthful language model should comprise (a) a perceptual learner that learns information about the world; (b) an extensional learner that maps sentences onto states of the world; and (c) an intensional learner which learns which sentences are synonymous. The paper further presents a toy algorithm that constrains LMs to only output truthful sentences.

As a disclaimer, I should say that this article is not related in any way to my areas of expertise, and I am surprised to see it assigned to me. I have never worked on formal semantics nor hallucination in LLMs.

**Questions For The Authors:**

[1] It would be very interesting to see some connections between your ideal truthful LM and contemporary NLP/AI models. How would a perceptual learner look like? Would it be something similar to a visual encoder? How would the extensional learning work? Is aligning sentences to states of the world somewhat akin to contrastive pre-training?

**Reasons To Accept:**

[1] Truth-theoretic semantics seems like a relevant and new approach to LLM hallucination.

[2] The proofs seem sound to me (but again, I am no formal semanticist).

[3] The paper is very well written.

[4] The overviews over the philosophical literature is accessible to non-philosophers.

**Reasons To Reject:**

[1] The paper formalizes some desiderata for truthful language models, but provide no concrete suggestions on how to implement them. For instance, consider the Learn-Babble-Prune algorithm. What the authors propose is that a model should verify its statements against its knowledge of the world and possible synonyms, and output a sentence only if it is true. The interesting thing would be how to implement this, which is not addressed in the paper.

[2] I feel there are some missed opportunities to connect this approach with contemporary NLP/AI work (see Questions).

**Reproducibility:**

N/A: Doesn't apply, since the paper does not include empirical results.

**Reviewer Confidence:**

1: Not my area, or paper was hard for me to understand. My evaluation is just an educated guess.

**Typos Grammar Style And Presentation Improvements:**

ll. 146-147 "that that"

ll. 218-219 "R : L → Ω, which map states of the world onto strings of the language". Here it seems that the direction is the opposite, from language to states of the world.

l. 222 "the pair ⟨ℓ, R(ℓ)⟩ is a sentence". In l. 144, ℓ alone was a sentence.

---

> ### Author Rebuttal · Authors · 2023-08-25
>
> Thank you for the comments and feedback! Your mention of visual encoders, and contrastive methods are very germane; thinking through the use of visual encoders in this context was particularly helpful. You raise a good point, and we stress that a wide variety of existing methods could be used to perform different tasks within this framework.
>
> By way of a response to your question, we propose:
>
> i) to include the section **Implementation examples** following $\S 4.3$. (We consider this to be in the spirit of `reporting additional experiments'.)
>
> ii) to include an abridged version of the below discussion of **multimodal encoders** and **contrastive methods** following the proposed $\S 4.4$, either in its own section, or in the section on related literature. (We consider this to be in the spirit of `adding additional references to the literature'.)
>
> ### **Applications of Learn-Babble-Prune**
>
> **Example 1: A factual text-to-text LLM**
>
> *Learn:*
>
>  1. Perceptual learning: We scrape an online encyclopedia, and designate this as our verified set of strings.
>  2. Intensional learning: We learn a set of paraphrases of the verified set of strings.
>
> *Babble:* An LLM generates a response to a query.
>
> *Prune:* The response is rejected if it is not a paraphrase of a sentence in the verified set of strings.
>
> **Example 2: A faithful image-to-text LLM**
>
> *Learn:*
>  1. Extensional: We pre-train a visual encoder, which learns a mapping from images (states of the world) to strings.
>  2. Perceptual: We designate a test set of images as ground truth about the environment. We apply the visual encoder to this test set of images.  We designate its output as our verified set of strings.
>  3. Intensional: We learn a set of paraphrases of the verified set of strings.
>
> *Babble:* An LLM generates a response to a query about the environment.
>
> *Prune:* The response is rejected if it is not a paraphrase of a sentence in the verified set of strings.
>
> Since the source domain is arbitrary, Example 2 covers a wide variety of use cases.
>
> [figures]
>
> Note that the text-to-text learning example is different to the world-to-text learning example, in that there is no explicit extensional learning in the former. There, the encoding of information about the external world has already been done -- by the authors of the encyclopedia entry.
>
> ### **Examples of extensional, perceptual, and intensional learners**
>
> **Multimodal encoders as extensional and perceptual learners**
>
> When we train a multimodal [any domain]-to-text encoder, we do extensional learning, because we learn a mapping from some source domain (i.e. states of the world) to text.
>
> When we apply the multimodal encoder to a ground-truth test set (observations of our environment), we are doing perceptual learning. The output strings are treated as our knowledge about the world.
>
> This connects to the literature on generalizable RL, especially in visual learning, where pre-trained encoders are used (e.g. Yuan et al. 2022, Radford et al, 2021, Tachet et al. 2018, Zhu et al. 2017). Similarly, approaches that use intermediate representations to improve generalization are closely related ([Chen et al. 2020](https://arxiv.org/abs/2011.06698)) This should not be entirely surprising, since the problem outlined in $\S 3$ is an example of distribution shift.
>
> **Contrastive similarity methods for intensional and extensional learning**
>
> Intensional case: take $\ell, \ell' \in L$ strings, and $\mathcal{L}$ a loss function. Then, learning $\mathcal{L}(\ell, \ell')$ is learning the similarity between strings. A paraphrase is an exact match; similarity is not synonymy, per the discussion above. But we might get synonymy when $\mathcal{L}(\ell, \ell') < \epsilon$, and contrastive pre-training could be used for intensional learning.
>
> Extensional case: take $\ell \in L$, $x \in X$, training observations in some non-text domain, and $y \in Y$ binary labels. Then learning the set of triples $(\ell, x, y) \:  \mathcal{L}(\ell, x, y) < \epsilon $ is extensional learning.
>
> **Paraphrase methods for intensional learning**
> A wide variety of methods for generating paraphrase corpora exist (e.g. Huang and Chang. 2021;  Egonmwan and Chali, 2019; Hu et al. 2019).

---

### Official Review · Reviewer_Leq6 · 2023-08-02

**Soundness:** 4

**Excitement:**

4: Strong: This paper deepens the understanding of some phenomenon or lowers the barriers to an existing research direction.

**Paper Topic And Main Contributions:**

This paper proposes a theoretical framework to understand why LLMs struggle to produce truthful outputs. This framework is based on an analytic take on the philosophy of language, and provides a theoretical procedure to devise systems that would systematically produce truthful outputs.

**Questions For The Authors:**

A/ Below are some elements of the mathematical exposition which I find troubling. Given the very tight deadlines for reviewing, I did not have enough time to fully work out all of the details of the proposal; nonetheless I would appreciate if the authors could clarify these:
- Definition 4.3 for an orbital seems to devolve into $X$ for all $x$ (given that all $g$'s are symmetries, all of their images should be exactly $X$). Maybe the authors meant the set $\\{g(x) | g \in G \\}$?
 - The definition of a paraphrase map $\pi$ lines 283 sq, seems problematic. You define it as "_a function $\pi$ that maps a string $\ell$ onto its paraphrases._"
This is not a function in the mathematical sense since one item is associated to multiple images, making the concept you introduce somewhat unwieldy. From a conceptual point of view I don't see how to construe paraphrasing as a function: all means of producing paraphrases need not apply to all sentences, and one sentence can have any number of paraphrases. I would suggest that a more natural way of conceiving paraphrases might be to rely on a partition of $L = \\pi_1 \cap \dots \cap \pi_n$ with added conditions; for instance such that $\forall \pi_i ~ \forall \ell_1, \ell_2 \in \pi_i \quad R(\ell_1) = R(\ell_2)$.
- You do nonetheless provide a more rigorous definition of paraphrase maps in definition 4.4. However, unless I'm mistaken, this definition is not consistent with claim 4.5:
  + we can construct a function $\pi_0$ that maps all strings $\ell$ unto a given constant string $\ell_0$, or formally $\forall \ell \in L ~ \pi_0(\ell) =  \ell_0$.
  + In such a scenario, we do have $\exists \ell ~ R(\ell) = R(\pi_0(\ell)$: by construction, we have $\pi_0(\ell_0) = \ell_0$ and thus $R(\pi_0(\ell_0)) = R(\ell_0)$, thereby satisfying definition 4.4
  + but $\pi_0$ is *not* a symmetry of $L$, since $\\{\pi_0(\ell) : \ell \in L \\} = \\{\ell_0 \\} \neq L$, contradicting definition 4.1
  + therefore claim 4.5 does not follow.
- Another degenerate case impacts claims 4.5 and 4.7 occurs if $R$ maps onto too few elements (e.g., taking $\forall \ell \in L \quad  R(\ell) := \omega_0$ )

**Reasons To Accept:**

- The paper develops a cogent theoretical framework and can provide novel insights on issues that are currently hotly debated in the NLP community. I believe it could generate interesting discussions.
- The approach is deeply rooted in analytical philosophy, and therefore draws an interesting connection between a current issue and a vast, underused literature. I strongly believe that such approaches should be fostered
- The paper proposes a (mostly theoretical) procedure to advance the field towards truthful LLMs, a worthwhile endeavour that the community would benefit from.

**Reasons To Reject:**

- I have some concerns with specific definitions and claims. See the "questions" box for details; as the article currently stands, I am uncertain that the framework is mathematically sound. At the very least, there seems to be a couple of crucial assumptions being left out the formalism.

- On a conceptual level, I am not certain that truthfulness necessarily encompasses everything that the community construes as hallucinations. Take the following two examples:
  + a  model generating an incoherent cooking recipe, by introducing ingredients that were not mentioned in the list, or listing incompatible steps (e.g., pureeing and then slicing the same ingredient). As (English) recipes are usually written as sequences of instructions, this complicates the derivation of a truth value for the models' productions&mdash;whereas the community might nonetheless frame this as a hallucinatory output.
  + a model can inject true, but entirely irrelevant strings that the community would construe as hallucinations. Consider for instance the following constructed example, which would be a strongly detached hallucination in the sense of [Guerreiro et al (2023)](https://aclanthology.org/2023.eacl-main.75/):
> The following contains an English translation of the French sentence "La tour Eiffel est grande." : "Donald Trump did not manage to close his umbrella, and the Eiffel tower is tall."

**Reproducibility:**

N/A: Doesn't apply, since the paper does not include empirical results.

**Reviewer Confidence:**

3: Pretty sure, but there's a chance I missed something. Although I have a good feel for this area in general, I did not carefully check the paper's details, e.g., the math, experimental design, or novelty.

**Typos Grammar Style And Presentation Improvements:**

- line 543: mode 4.10 is strictly truthful, should be model 4.9
- equation 4, Theorem 3.3, Model 4.9 should not continue on past the margins.

---

> ### Author Rebuttal · Authors · 2023-08-25
>
> Thank you for the above comments, which are very helpful! We appreciate close attention to the math.
>
>
> **Technical points on $\S 4.1$**
>
> - {$g(x) | g \in G$} was indeed the intended definition.
>
>  - The informal version of the definition of a paraphrase map should read ``a bijective function $\pi$ that maps strings onto paraphrases.'' Likewise Definition 4.4 should specify that $\pi$ is bijective, and the quantifier should be $\forall$ rather than $\exists$. (For expository purposes, there is actually no need to introduce $\pi$ before $\S 4$; we will delete the informal introduction of paraphrase maps on p.4, and edit Proposition 2.5 so that the second condition is stated in terms of R instead.)
>
>   - We do not intend to motivate $\pi$ as a correspondence. We are interested in the collection of bijective paraphrase maps $\Pi$.
>
>      - Intuitively: $\pi$ is a permutation of L that swaps only strings that have the same referent (meaning).
>
>      - Constructing the collection of paraphrase maps $\Pi$ from paraphrases $\pi$ is then essentially the same thing as constructing the set of all permutations (hence $\Pi$ and $\pi$).
>
>      - The natural generating set here is the set of two-place semantic substitutions; i.e. swapping any two strings in a list that have the same referent. The collection of all such two-place substitutions is the set of all (semantic) permutations, aka paraphrases.
>
>      - The conclusion that $\Pi$ is a symmetry group of $L$ is then essentially the same claim that the set of all permutations of $X$ is the symmetric group of $X$.
>
> - Two additional assumptions were included in an earlier version of this paper (and we propose adding these back in). We believe these rule out the degenerate cases discussed. Every string has exactly one referent, and that there is at least one referent. Specifically:
>
>      - A language $L$ is fully-interpreted if: $\forall \ell \in L, \exists ! \omega \in \Omega$ such that $R(\ell) = \omega$.
>
>      - $\Omega$ is non-empty.
>
> - Since $\pi$ is a bijection, we start with the full list of words, and every substitution of words with the same meaning returns the same list of words. (This is an informal proof of Claim 4.5.)
>
> - Once the map $R$ is determined, $\Pi$ is determined, so given a fully-interpreted language, the collection of acceptable transpositions is given, not something that we can construct however we like.
>
> - With this context specified, the proof of Claim 4.5 is very simple (and we propose including this).
>
> - Since each $\pi \in \Pi : L \to L$ is a bijection, and hence is a permutation of the elements of $L$, we have that, $\forall \pi \in \Pi$: {$\pi(\ell) : \ell \in L$}$ = L$. Hence $\Pi$ is a symmetry group of the set $L$.
>
> - We propose Claim 4.6 and $\S 4.1.2$ should be replaced by:
>
>      - Definition 4.6: The orbit of $\ell \in L$ under $\Pi$ is the set $I(\ell) =$ {$\pi(\ell) | \pi \in \Pi$}.
>
> - This cleans up notation, since $\pi$ is already defined in terms of R, and allows us to express the second line of model 4.9 as:
>
> $$Pr(\ell \in I(\ell))Pr(I(\ell) | \omega)Pr(\omega) $$
>
> which is simpler. Further $\S 4.1.2$ is not correct as written. We do not need a symmetry group account of extension: we just need a learnable map from $L \to \Omega$. (Thinking about visual encoders clarified this point.)
>
> - Lastly, the notational changes in Model 4.9 are carried through to Algorithm 1.
>
> - Notice that the collection of unique orbits does, in fact, define a partition of $L$, as per the reviewer's proposal.
>
> **The scope of the term hallucination**
>
> This point is well-taken.
>
> Note that definition 3.1 defines (strictly) truthful LLMs, which would be a factual LLM.  We omitted from this draft the definition of a (conditionally) truthful LLM, which would be a faithful LLM.
>
> This paper was originally titled 'Towards Truthful LLMs'. Perhaps 'Towards factual and faithful LLMs' would be more appropriate as the kicker in the title.
>
> We would be happy to add a paragraph providing additional clarification on this point, and to contextualize the scope of the contribution as directly bearing on a subset of types of hallucination. We are also open to opting for this alternate title, if ``hallucinate'' is considered misleading or overly broad.

---

### Official Review · Reviewer_L6mo · 2023-08-05

**Soundness:** 3

**Excitement:**

4: Strong: This paper deepens the understanding of some phenomenon or lowers the barriers to an existing research direction.

**Paper Topic And Main Contributions:**

The paper explores the challenges faced by truthful Language Learning Models (LLMs) and the identification problem of generating accurate sentences beyond the training data, leading to hallucination. The authors identify three essential tasks for a truthful LLM: perceptual learning, extensional learning, and intensional learning. They show that failure to perform these tasks can lead to hallucination, even with training on a factual learner-generated corpus. Additionally, the paper proposes the "Learn-Babble-Prune" procedure to ensure strictly truthful outputs from LLMs by employing rejection sampling in the presence of consistent perceptual, extensional, and intensional learners.

**Reasons To Accept:**

This paper addresses a crucial and timely issue in the field of Language Learning Models (LLMs) – the problem of systematic hallucination. By providing a new formalization and drawing upon concepts from analytic philosophy of language, the authors offer valuable insights into the inherent challenges faced by LLMs in maintaining truthfulness. The introduction of the model comprising three essential learners – perceptual, extensional, and intensional – presents a novel approach to generate strictly truthful outputs. The proposed "Learn-Babble-Prune" procedure serves as a heuristic example of how to achieve this goal, guiding researchers towards conceptual clarity and highlighting the significance of addressing hallucination in language models. Overall, the paper's contributions have the potential to enhance the reliability and credibility of future LLMs.

**Reasons To Reject:**

- Lack of empirical evidence: The paper lacks concrete experimental validation, weakening the credibility of its claims and proposed solutions.

- Overreliance on abstract concepts: The heavy use of analytic philosophy of language makes the paper less accessible to a broader audience in the natural language processing community.

- Inadequate practical demonstration: The proposed "Learn-Babble-Prune" procedure remains a heuristic example, with unclear feasibility and applicability, hindering its real-world impact.

**Reproducibility:**

3: Could reproduce the results with some difficulty. The settings of parameters are underspecified or subjectively determined; the training/evaluation data are not widely available.

**Reviewer Confidence:**

2: Willing to defend my evaluation, but it is fairly likely that I missed some details, didn't understand some central points, or can't be sure about the novelty of the work.

---

> ### Author Rebuttal · Authors · 2023-08-24
>
> Thank you for the comments and feedback! We have been thinking about how to make the proposed approach more concrete and useful for applied researchers and practitioners.
>
> We propose:
>
> i) to include the section **Implementation examples** following $\S 4.3$. (We consider this to be in the spirit of `reporting additional experiments'.)
>
> ii) to include an abridged version of the below discussion of **multimodal encoders** and **contrastive methods** following the proposed $\S 4.4$, either in its own section, or in the section on related literature. (We consider this to be in the spirit of `adding additional references to the literature'.)
>
> ### **Applications of Learn-Babble-Prune**
>
> **Example 1: A factual text-to-text LLM**
>
> *Learn:*
>
>  1. Perceptual learning: We scrape an online encyclopedia, and designate this as our verified set of strings.
>  2. Intensional learning: We learn a set of paraphrases of the verified set of strings.
>
> *Babble:* An LLM generates a response to a query.
>
> *Prune:* The response is rejected if it is not a paraphrase of a sentence in the verified set of strings.
>
> **Example 2: A faithful image-to-text LLM**
>
> *Learn:*
>  1. Extensional: We pre-train a visual encoder, which learns a mapping from images (states of the world) to strings.
>  2. Perceptual: We designate a test set of images as ground truth about the environment. We apply the visual encoder to this test set of images.  We designate its output as our verified set of strings.
>  3. Intensional: We learn a set of paraphrases of the verified set of strings.
>
> *Babble:* An LLM generates a response to a query about the environment.
>
> *Prune:* The response is rejected if it is not a paraphrase of a sentence in the verified set of strings.
>
> Since the source domain is arbitrary, Example 2 covers a wide variety of use cases.
>
> [figures]
>
> Note that the text-to-text learning example is different to the world-to-text learning example, in that there is no explicit extensional learning in the former. There, the encoding of information about the external world has already been done -- by the authors of the encyclopedia entry.
>
> ### **Examples of extensional, perceptual, and intensional learners**
>
> **Multimodal encoders as extensional and perceptual learners**
>
> When we train a multimodal [any domain]-to-text encoder, we do extensional learning, because we learn a mapping from some source domain (i.e. states of the world) to text.
>
> When we apply the multimodal encoder to a ground-truth test set (observations of our environment), we are doing perceptual learning. The output strings are treated as our knowledge about the world.
>
> This connects to the literature on generalizable RL, especially in visual learning, where pre-trained encoders are used (e.g. Yuan et al. 2022, Radford et al, 2021, Tachet et al. 2018, Zhu et al. 2017). Similarly, approaches that use intermediate representations to improve generalization are closely related ([Chen et al. 2020](https://arxiv.org/abs/2011.06698)) This should not be entirely surprising, since the problem outlined in $\S 3$ is an example of distribution shift.
>
> **Contrastive similarity methods for intensional and extensional learning**
>
> Intensional case: take $\ell, \ell' \in L$ strings, and $\mathcal{L}$ a loss function. Then, learning $\mathcal{L}(\ell, \ell')$ is learning the similarity between strings. A paraphrase is an exact match; similarity is not synonymy, per the discussion above. But we might get synonymy when $\mathcal{L}(\ell, \ell') < \epsilon$, and contrastive pre-training could be used for intensional learning.
>
> Extensional case: take $\ell \in L$, $x \in X$, training observations in some non-text domain, and $y \in Y$ binary labels. Then learning the set of triples $(\ell, x, y) \:  \mathcal{L}(\ell, x, y) < \epsilon $ is extensional learning.
>
> **Paraphrase methods for intensional learning**
> A wide variety of methods for generating paraphrase corpora exist (e.g. Huang and Chang. 2021;  Egonmwan and Chali, 2019; Hu et al. 2019).

---

### Meta-Review · Area_Chair_wsX3 · 2023-09-15

**Recommendation:** 5

**Metareview:**

Reviews provide scores of 3,4,4 for soundness and 4,4,4 for excitement.

Condensing the reviews, strengths and weaknesses including the following were mentioned.

Strengths:

- addresses a crucial and timely issue for the field (R1, R2, R3)
- interesting links to philosophy (R1, R2)

Weaknesses:

- no empirical evidence or validation (R1, R3)
- high degree of abstraction limits accessibility (R1)
- questions about mathematical soundness were originally raised by (R2), but their final soundness score is 4

---

### Decision · Program_Chairs · 2023-10-07

**Decision:**

Accept-Main

**Comment:**

Reviews provide scores of 3,4,4 for soundness and 4,4,4 for excitement.

Condensing the reviews, strengths and weaknesses including the following were mentioned.

Strengths:

- addresses a crucial and timely issue for the field (R1, R2, R3)
- interesting links to philosophy (R1, R2)

Weaknesses:

- no empirical evidence or validation (R1, R3)
- high degree of abstraction limits accessibility (R1)
- questions about mathematical soundness were originally raised by (R2), but their final soundness score is 4